# The Race for Global Equitable Access to COVID-19 Vaccines

**DOI:** 10.3390/vaccines10081306

**Published:** 2022-08-12

**Authors:** Lukman Nul Hakim Md Khairi, Mathumalar Loganathan Fahrni, Antonio Ivan Lazzarino

**Affiliations:** 1Faculty of Pharmacy, MARA, University of Technology (UiTM), Selangor Branch, Puncak Alam Campus, Puncak Alam 42300, Malaysia; 2Pharmacy Department, Hospital Sultanah Nur Zahirah, Ministry of Health Malaysia, Kuala Terengganu 20400, Malaysia; 3Collaborative Drug Discovery Research (CDDR) Group, Communities of Research (Pharmaceutical and Life Sciences), Universiti Teknologi MARA (UiTM), Puncak Alam 42300, Malaysia; 4Department of Epidemiology and Biostatistics, School of Public Health, Imperial College London, London W2 1NY, UK

**Keywords:** SARS-CoV-2, COVID-19 vaccines, health service accessibility, global health, health equity, immunization programs, COVID-19, inoculation, low- and middle-income countries

## Abstract

COVID-19 vaccines are possibly the most effective medical countermeasures to mitigate and ultimately bring to a halt the COVID-19 pandemic. As we transition to endemicity, inequitable access to vaccines, and particularly in low- and middle-income countries (LMICs), still poses risks of unprecedented disruptions and the emergence of viral mutations, which potentially lead to notorious vaccine-resistant variants. The missteps learned from the previous responses to the human immunodeficiency virus (HIV) and influenza outbreaks founded the hypothetical plan to ensure that vaccine accessibility to LMICs is not impeded. The SARS-CoV-2 vaccines’ social promise was to lessen the underlying racial, ethnic, and geographic inequities that COVID-19 has both made apparent and intensified. Vaccine nationalism was evident throughout the COVID-19 crisis. Many high-income countries directly negotiated large advance orders for the vaccines, leaving resource-limited countries scrambling for access. This occurred despite international initiatives to structure the development and equitable distribution of vaccines, channeled through a vaccine pillar: COVID-19 Vaccines Global Access (COVAX). The serious supply shortages and national procurement methods of some countries that bypassed the vaccine pillar hindered the optimal function of COVAX in delivering timely and adequate doses to participating countries. COVAX strategized its approach by promoting fundraising, coordinating vaccine donations from countries with surplus doses, expediting reviews of vaccine candidates, and facilitating the expansion of the manufacturing capacity. While increasing capacity for production, technology transfer led to lesser siloes, enhanced manufacturing standardization, and less secrecy over production data. Ultracold storage requirements for leading vaccines were a considerable hurdle to the global immunization efforts, and particularly in LMICs with limited equipment and resources to support sophisticated cold-chain systems. Manufacturers strived to ease cold-chain restrictions on the basis of stability data submitted to national regulatory bodies. The development of single-dose vaccines offered promising solutions to simplify the administrative and logistic complexities that existed within the COVID-19 vaccination programs. As such, the requirements for both ultracold storage conditions were eased, and concerns over booster doses were addressed. To expand coverage, the dosing intervals of the Oxford/AstraZeneca vaccines were extended according to data from Phase III clinical trials on effectiveness. In addition, with the recent outbreak of monkeypox, the lessons from past experiences of curbing infectious diseases, including COVID-19, must be learned and acted upon. The review summarizes the global efforts with respect to vaccine development, production, allocation, and deployment to achieve equitable access.

## 1. Introduction

The adoption of new norms amidst the transition to the COVID-19 endemic phase is timely and welcomed. More than two years since the World Health Organization (WHO) declared the pandemic, the world has witnessed an unimaginable scale of calamity and daily new confirmed COVID-19 deaths per million people (2122 deaths on 1 June 2022) [1]. The severe acute respiratory syndrome coronavirus 2 (SARS-CoV-2) was the stimulus for the COVID-19 crisis. Its genome encodes the surface-exposed structure of spike (S) glycoproteins. Their important roles in viral attachment, fusion, and entry into the host cell predisposed the S glycoproteins to be a direct target for host immune responses and the neutralization of antibodies, and, subsequently, they became the focus of most vaccines and therapeutic approaches [2].

Similar to other RNA viruses, SARS-CoV-2 undergoes rapid evolution through the accumulation of mutations within its viral genome, which has led to the emergence of variants of concern (VOC) from India (B.1.617.2), South Africa (B.1.351), the United Kingdom (B.1.1.7), and Brazil (P.1). The occurrence of VOC calls for a more stringent and consolidated approach to control the spread, as they are associated with higher transmissibility and the risk of mortality [3]. Despite the emergence of new variants, the rollout of vaccines initiated by Pfizer on 8 December 2020 offered a promising path to the attainment of herd immunity. This was viewed as key to bring to a halt the pandemic [1].

Upon testing in Phase III clinical trials, and now with pharmacovigilance and the post marketing surveillance of vaccines, the US Food and Drug Administration (FDA) and the WHO actively monitor the incidences of adverse events following immunization (AEFI) [1,4]. Other vaccines, including Moderna, Oxford/AstraZeneca, Janssen, Sinovac, Sputnik V, Sinopharm, CanSino, EpiVacCorona, Covaxin, RBD-Dimer, Abdala, and Soberana02, have been approved for emergency use in numerous countries [4]. In the United States, the FDA issued initial emergency use authorization (EUA) for Pfizer/BioNTech on 11 December 2020, and for Moderna COVID-19 vaccines on 18 December 2020. Of note, both vaccines were developed at a relatively accelerated rate using the novel mRNA technology, and they assisted the early vaccination efforts, which led to reduced hospitalization and mortality rates. These vaccines were subsequently granted full approval by the FDA in August 2021 (Pfizer/BioNTech) and January 2022 (Moderna) [5]. The EUA for the Pfizer/BioNTech vaccine was amended to include adolescents from 12 to 15 years of age (issued on 10 May 2021), and children from 5 to 11 years of age (issued on 29 October 2021), which allowed extended coverage to the pediatric population (Figure 1). Approval by national regulators furnished some degree of confidence for individual countries in the vaccines’ safety, quality, and efficacy, thereby rendering them suitable for administration [4].

## 2. Lessons Learned from Previous Infectious Disease Outbreaks

The missteps learned from the previous responses to the human immunodeficiency virus (HIV) and influenza outbreaks founded the hypothetical global plan to ensure that vaccine accessibility to LMICs is not impeded. The SARS-CoV-2 vaccines’ social promise was to lessen the underlying racial, ethnic, and geographic inequities that COVID-19 has both made apparent and intensified. Evidently, vaccine nationalism was at large throughout the COVID-19 crisis [7]. The pre-existing vulnerable economies of the LMICs meant that their citizens were already under a larger threat of unprecedented health, work, and livelihood disruption [4]. Thus, during the pandemic, there was and still is a dire need for global equitable access to safe and effective vaccines to sustain public health systems and ultimately overcome economic downturn, particularly for LMICs.

In the past, throughout the HIV and influenza outbreaks, the world witnessed disadvantaged nations, and particularly Africa, struggle with access to life-saving therapies. In those trying times, as antiretroviral drugs for HIV treatment began to penetrate the global market in the mid-1990s, the initial costs and prices of the drugs skyrocketed, which made the drugs hardly affordable [7]. While AIDS-related mortalities in rich and developed countries sharply declined, the death of over 12 million Africans between 1997 and 2007 was attributable to the inaccessibility to the ground-breaking drugs [7]. The influx of drugs to the continent were seen only in the early 21st century, following large funding commitments by the United States President’s Emergency Plan for AIDS Relief (PEPFAR) and the Global Fund to Fight AIDS, Tuberculosis and Malaria [8,9].

International calls to redefine a global access framework for epidemic vaccines date back to the year 2004, when highly pathogenic avian influenza A (H5N1) (HPAI-H5N1) re-emerged. Amidst the worldwide fear of a potential pandemic, frantic plans for H5N1 vaccine research and development commenced [10]. However, the plans received negative responses from developing countries due to overwhelming concerns that, because of high patent prices, those vaccines would not be readily accessible to their populations. In 2007, Indonesia was one of the countries that was hit hard by avian influenza. However, a request by the WHO for H5N1 virus samples was declined and, as such, the surveillance and vaccine development efforts were hindered. In the absence of a global access framework for equitable vaccine distribution, fears mounted that the countries in dire need, or in which prevalence and incidences of outbreaks were high, were not actually receiving the required treatment [11]. The WHO subsequently attempted to initiate negotiations for the sharing and stockpiling of H5N1 vaccine doses, but to no avail [12].

When another influenza outbreak of a different strain (H1N1) shocked the world in 2009, vaccines were considered of paramount importance against a highly infectious strain that spread infections easily and quickly around the globe [13]. Access inequity was apparent as more developed countries swept up sizeable vaccine orders in advance, procuring nearly all of the manufactured doses [14]. Despite their initial promises to donate parts of the vaccines to LMICs under the WHO and United Nations-sponsored plans, their nationalistic behavior overrode, and thus the premediated planning and strategizing were futile, and the distribution mechanism proved flawed. In fact, the more affluent countries proceeded to secure supplies for their own populations, leaving LMIC countries with limited supplies and the untimely delivery of vaccines [15]. The H1N1 vaccine distribution inevitably occurred on the basis of the purchasing power of the high-income countries and was not based on the geographical justification for the epidemiology or risk of transmission.

## 3. Global Allocation

### 3.1. The Issue of Vaccine Nationalism

In mid-2010, the Ebola epidemic in West Africa unearthed the lack of international capacity to be collectively prepared and responsive, and to recover from outbreaks of infectious diseases. The unpreparedness in managing an epidemic crisis prompted recommendations by the United Nations Secretary-General’s Global Health Crises Task Force for the establishment of an independent monitoring and evaluation body to safeguard the global interest in and preparedness for health calamities. This led to the launch of the Global Preparedness Monitoring Board (GPMB) in 2018. Within a year, the GPMB produced its inaugural report titled “A World at Risk”, which cautioned against the risk of rapidly spreading respiratory pathogens owing to the knowledge of the intermittent yet unpredictable and abrupt emergence of new influenza strains [16]. In the report, recommendations were made for countries to harmonize the “means to share limited countermeasures across countries” [16]. Merely three months later, world leaders were once again faced with a health emergency. This time, the catastrophes struck with far greater magnitude. COVID-19 destabilized global healthcare frontiers, toppled finances, and redefined the very essence of social norms.

The GPMB’s preconceived notion that countries could be deviating from the consensus and international concerted efforts to achieve equitable access to scarce medical countermeasures became a reality when national leaders opted for “vaccine nationalism” in the wake of the COVID-19 pandemic [17]. Many high-income countries with substantial resources directly negotiated huge advance orders for the vaccine in return for considerable investments in research and production infrastructure. For instance, the United Kingdom was the first country in the world to attain access to AstraZeneca’s vaccines, which were codeveloped with the University of Oxford. The first 30 million doses that were manufactured were allotted in exchange for an investment worth USD 79 million [18]. Additionally, AstraZeneca reported that the United States was promised a delivery of at least 300 million doses as early as October 2020, and this was after the country pledged USD 1.2 billion worth of investment in its company [19]. One has to recognize that, to bring COVID-19 crises to a complete halt, or at least to states of endemicity, the medical countermeasures ought to reach every nook and cranny. Because of its high rate and strength of infectivity, the entire population living across the globe requires full immunization. This should remain the aim, regardless of a nation’s financial strength and resources to secure the valuable supplies.

The inequitable deployment of vaccines in LMICs poses the risk of the divergent quality of healthcare, economic-shock-induced global inequities, large-scale migration, and insecurity [20]. A further deterioration in global health and economic calamity could occur should high-income nations continue to opt for a nationalistic stance. With the latest outbreak of monkeypox, lessons from past experiences of managing infectious diseases, and particularly COVID-19, must be learned and acted upon [21,22].

### 3.2. Overcoming Vaccine Nationalism through COVAX

A “Call to Action for the Access to COVID-19 Tools (ACT) Accelerator (ACT-A)” was issued on 24 April 2020 by the President of the European Commission, the Director-General of the WHO, heads of states, and other development partners. This was followed by a pledge for international solidarity by various countries, civil society, industries, and development partners in two consecutive events [23]. The ACT-A is a global coordinating mechanism that is targeted at boosting the development and equitable distribution of three important countermeasures: therapeutics, diagnostics, and vaccines, through actions from multiple international stakeholders.

Structured initiatives for vaccines are channeled through the vaccine pillar, named COVID-19 Vaccines Global Access (COVAX), which was co-led by the Coalition for Epidemic Preparedness Innovations and the Global Alliance for Vaccines and Immunizations (Gavi). COVAX aims to support the timely research and development of effective vaccines to be used in all countries. Instead of relying on an uncoordinated method in which countries scramble to secure vaccine doses directly from the manufacturers for their own population use, a proportional distribution scheme through COVAX is theoretically better organized and reasonable. This requires cooperation and commitment at the international level to prevent the hoarding of vaccines [24].

To secure two billion vaccine doses for fair distribution by the end of 2021, and particularly in LMICs, the initiative strived to strengthen the manufacturing infrastructure and procurement capabilities [23]. The Coalition for Epidemic Preparedness Innovations (CEPI) is the leading foundation in acquiring strategic vaccine development partnerships with nations around the world, building the capacity needed for vaccine research and development, and reinforcing local expertise [25]. Meanwhile, the Global Alliance for Vaccines and Immunization (Gavi) is an international public–private partnership that focuses on the procurement and distribution of COVAX through its suborganizations—the COVAX Facility and Gavi COVAX Advance Market Commitment (AMC) [26]. CEPI and Gavi work alongside the WHO and its vaccine review body of the Strategic Advisory Group of Experts on Immunization (SAGE) Committee, which developed the values framework for the allocation and prioritization of COVID-19 vaccination [27]. The values framework advocates that vaccine distribution shall consider the epidemic risks and necessities of all nations, with special attention given to LMICs. The pandemic presents with a global recession, which is the first since World War II. Thus, the WHO SAGE Committee recommends that high-income countries abstain from undercutting vaccine supplies to nations with less political or economic power, and instead participate in achieving global access through COVAX [27,28].

### 3.3. Is Inequitable Allocation Nonexistent with COVAX?

More than 700 million doses of varying vaccine brands were administered in over 194 countries within four months of the global rollout in December 2020 [29]. Fourteen countries had yet to start vaccinations due to the procurement challenges posed beyond the COVAX Facility. They were not ready, nor were they planning to begin within the next couple of weeks or months. However, the WHO stated that 87% of these doses had reached high-income and upper-middle-income countries, while the lower-income countries only received a dismal proportion of 0.2% [30]. An apparent inequity in the global vaccine distribution was indicated, as 1 in every 4 people in wealthy countries had the vaccine, as opposed to merely 1 in an excess of 500 individuals in less fortunate countries. It was even more alarming when, simultaneously, physicians, who were a minority and a scarce subpopulation in sub-Saharan Africa, died from COVID-19 due to not receiving the rightful protection and immunization [31]. The WHO attributed the supply shortage to COVAX, citing allocation disparity as the main reason for the inequity. By the end of March 2021, COVAX was originally expected to deliver up to 100 million doses, but the actual number had shrunk to only 38 million [30]. Moreover, the national procurement strategies of some countries that bypassed COVAX to directly deal with the vaccine developers based on commercial and political grounds posed another threat to global equity [32]. The practice of direct purchasing that was initiated by high-income countries was emulated by the rest of the world. As of February 2021, 62 nations had taken matters into their own hands, initiated procurement agreements, and signed off on deals directly with manufacturing companies [33].

Nevertheless, COVAX was of relevance to countries that were not financially able to self-procure sufficient doses, and in particular, the low-income countries. Agreements reached by COVAX with five manufacturers to secure 2 billion doses were in line with the organizational goal of inoculating at least 20% of its members’ populations by the end of 2021 [32,34]. Many other strategies enforced by COVAX evolved around efforts to speed up production and supply. The alliance sought vaccine donations from high-income countries with surplus doses, and it expedited the review of vaccine candidates and the expansion of the global manufacturing capacity [30]. The success of COVAX highly depended on the availability of financial resources to procure vaccines. As of May 2021, the financial donations amounted to USD 6.3 billion and originated mainly from government agencies and partners. To further bridge the gap of USD 2.6 billion, the WHO Foundation launched a mass fundraising campaign called “Go Give One”, calling for small contributions from everyone around the world [35,36]. Many wondered whether or not COVAX’s target of equity in the early stages of global immunization was too farfetched. In response to this, COVAX provided its assurance that, with monetary and external support, it could effectively accelerate access to vaccines for lower-income countries through provision at reduced prices, and the assistance provided by initiating vaccination programs at a much sooner rather than later date [32].

By January 2022, the global inequities in the distribution of COVID-19 vaccines were better alleviated through internationally pooled procurement within the COVAX Facility. Poorer countries received greater benefits than higher-income nations from the numbers of COVAX-secured vaccine doses [37]. The largest disparities between the shares of total allocated and distributed vaccine doses were still notably apparent in countries with lower income levels. However, this was due to a lack of cold-chain readiness rather than inequality in distribution [32]. To assist COVAX-eligible countries to prepare for meeting the minimum conditions of the vaccine distribution system, the WHO and the United Nations Children’s Fund (UNICEF) established a guidance note on the logistics and supply management of COVID-19 vaccines [38].

## 4. Vaccine Development and Manufacture

### 4.1. The Development of Single-Dose COVID Vaccines

The availability of single-dose vaccines offers a promising solution to streamlining the administrative challenges that are faced during inoculation efforts. Unlike the mRNA vaccines of Pfizer/BioNTech and Moderna, which require two doses and storage at subzero temperatures, the one-shot vaccine developed by Johnson & Johnson (J&J), named the Janssen COVID-19 vaccine, can be stored for a maximum duration of three months in a standard refrigerator from 2 °C to 8 °C [39].

In the same month (February 2021) that J&J received the greenlight from the FDA, another single-dose vaccine produced by China, named CanSino, was approved in its country of origin. The recombinant J&J and CanSino vaccines utilize a distinct manufacturing technique with adenovirus vectors type 26 and 5, respectively [40]. The technology provides an advantage for logistics in that distribution can occur at the normal refrigeration temperature, which is more cost saving and convenient than the requirements for the preceding vaccines, which needed to be transported frozen [41]. A one-shot vaccine regimen could potentially enhance the outreach to more remote areas, where the challenge usually lies in delivering the supplies without any degradation in the potency or safety. In addition, the setup for appointments can be conducted once, instead of the usual follow-up arrangement for booster doses, which ideally takes place with the preceding vaccination [42]. Moreover, an exploratory subgroup analysis of the Janssen vaccine for a Phase III trial in the United States found that the one-shot vaccine was 72% (95% confidence interval (CI), 58.2–81.7%) effective at preventing a moderate to severe/critical COVID-19 infection for at least 28 days following immunization [43]. Different endpoints were utilized for the Phase III trials of the CanSino vaccine, in which it was suggested that the vaccine was 68.83% and 65.28% effective in preventing symptomatic COVID-19 infection 14 days and 28 days, respectively, after administration [44]. Therefore, the development of single-dose vaccines, which are proven safe and effective, may assist in achieving global equitable access to the valuable countermeasures.

### 4.2. Technology Transfer as a Means to Boost the Manufacturing Capacity

In early 2021, countries of the European Union encountered severe supply shortages, which induced public frustration. Mass media and mass communications were used to exert pressure on political leaders [45]. Many LMICs are still facing far greater challenges in securing vaccine supplies. With the development and authorization of increasing numbers of vaccines for emergency use, government bodies and production companies need innovative solutions to prepare for imminent mass production and to meet the unprecedented global demand for vaccines. Failing this, and given the current and forecasted global recession for millions living across the globe, and particularly for those living in poorer countries, receiving their first jabs by 2024 might not be attainable [46].

The solution of relaxing patent protection is likely to invite much controversy in the profit-driven pharmaceutical industry. However, patent waivers in a time of epidemic crisis can speed up global vaccine access [47]. An alternative method to accelerate this is through technology transfer, in which the developer of the original pharmaceutical product forms partnerships with companies in middle-income countries. This facilitates the production of vaccines locally in distant regions. The Serum Institute of India, the world’s largest vaccine manufacturer, had such agreements with the developers of COVID-19 vaccines, namely, AstraZeneca and Novavax. AstraZeneca also made multiple collaborations for technology transfer with mAbxience Buenos Aires in Argentina, Fiocruz in Brazil, and Siam Bioscience in Thailand [32]. The nature of these partnerships is generally uncommon and is, in fact, unfavorable to the respective parent companies; however, a deal made between two of the world’s largest pharmaceutical manufacturers, Merck and Johnson & Johnson, to combine technologies for increased production was highly commendable [48]. From another perspective, technology transfer will not only be beneficial during the pandemic, but it could also potentially lead to fewer siloes, thereby enhancing the manufacturing quality through standardization, and deriving less profit-driven secrecy over production data in the future [49].

## 5. Vaccine Deployment

### 5.1. Vaccine Cold Chain Challenge

Maintaining the vaccine supply chain at constant temperatures is one of the biggest challenges posed to the deployment of temperature-sensitive products, including COVID-19 vaccines. A number of first-generation vaccines incorporated ultracold-chain technology and, hence, they possessed short-term stability outside the recommended storage conditions [50]. This inflicted substantial hurdles to the global immunization efforts, and particularly in low-income countries, which lack established equipment to support sophisticated cold-chain systems. Around one-fifth of the world’s most impoverished nations were not equipped with sufficient cold-chain capacities. In some other countries, despite having the infrastructure, the equipment was either broken or outdated and, as such, these nations struggled to sustain optimal temperatures for vaccines [51]. The Gavi Board recently pledged to explore its Cold Chain Equipment Optimization Platform (CCEOP) to potentially improve the cold-chain capacities in LMICs. One of the ways proposed is through the procurement of additional refrigerators, including solar-powered refrigerators, that are suitable for vaccines [52].

Meanwhile, the manufacturers, by way of improving formulations, also strived to ease their ultracold-chain requirements and extend shelf-lives to facilitate simpler administration and storage conditions. The Pfizer/BioNTech vaccine initially required administration within 5 days of removal from its ultralow freezing temperature (from −80 °C to −60 °C) [32]. On 25 February 2021, the drug company’s application was approved by the FDA, and an announcement was made that the newer vaccine batches would no longer be subjected to rigid cold-chain restrictions. Based on Pfizer Inc.’s submitted stability data to the national regulatory body, conventional temperatures that are normally found in pharmaceutical freezers would be permitted for the transportation and storage of undiluted frozen vials for a maximum period of 2 weeks [53]. This duration was in addition to the 5 days that was applicable to the storage of the thawed vials in the refrigerator [54]. The updated conditions required for transportation and storage greatly reduced the burden of purchasing ultracold-chain storage infrastructure at vaccination sites.

Similarly, Moderna, with the FDA’s approval, announced in early April 2021 that updates had been made to their regulations to allow their vaccine shots to be administered into the arms more efficiently and rapidly. The first revision to their regulations allowed the extension of the vials’ shelf-life after the first puncture from 6 h to 12 h [55]. The preceding shorter duration of stability evidently caused healthcare workers to hurry to administer before the 6 h shelf-life ended. On many occasions, young healthy individuals ended up receiving jabs despite not being listed among the national prioritization groups, as the leftover doses would have otherwise gone to waste [56]. Additionally, Moderna’s change in shipment sizes, to allow five additional doses per vial, was intended to facilitate mass vaccination through increased dose delivery [57]. Table 1 summarizes the latest attributes and requirements for the storage and handling of the main COVID-19 vaccines as of 1 July 2022.

### 5.2. Dosing-Interval Prolongation to Expand Coverage

The decision on the dosing intervals of COVID-19 vaccines was theoretically made based on the clinical-trial data submitted to the WHO and respective regulatory bodies [64]. The leading vaccines typically have defined durations of intervals between two doses for administration. Overwhelming concerns about the limited supply and the issue of vaccine logistics generated a need for the continuous revision of the dosing intervals. In view of the rapidly changing waves of infection, and with an intention to optimize the numbers to receive a first dose, several countries, including the United Kingdom, Canada, India, and Malaysia, implemented an extension to the schedule for the administration of the second dose [64,65,66]. This led to the question of whether or not the expanded vaccination coverage in the current environment of highly active transmissions might be offset by a hypothetical augmented risk for viral mutations and VOC [64]. Nonetheless, the preliminary data indicated that single-dose COVID-19 vaccines were capable of delivering high efficacy levels, thus supporting the motion for second-dose delays. This was substantiated by the Joint Committee on Vaccination and Immunization’s data analysis, which indicated that a single shot of either vaccine would be able to provide significant protection from within 2 to 3 weeks of inoculation for clinical diseases, and particularly for the severe presentation of COVID-19 diseases [67]. Despite being considered essential to ensuring durable protection, the expected increase in vaccine efficacy that was to result from the booster dose was likely to be rather “modest”.

In addition, effectiveness data from the United Kingdom and Canada, along with the mathematical modeling from respective clinical-trial publications, demonstrated that a single dose of the Pfizer/BioNTech or Oxford/AstraZeneca vaccines might offer protection for a prolonged period of 8 weeks from vaccination [68,69,70]. The extended interval between two doses of the Oxford/AstraZeneca vaccine was associated with an even higher effectiveness of the vaccine. Recent data from Phase III clinical trials that assessed the optional timing of the second dose found that the vaccine efficacy in the first 12 weeks did not wane. Participants who had their second jabs beyond 12 weeks of the first dose had 82.4% efficacy, as opposed to 54.9% in those who had an interdose interval of less than 6 weeks apart [64]. These findings lent support for inoculation approaches that delay the booster dose beyond the initial recommended interval of 4 weeks for the Oxford/AstraZeneca vaccine. This method received a subsequent endorsement by the WHO SAGE Committee [71]. Despite the limited ability to extrapolate the findings to other COVID-19 vaccines, these early findings substantiate the need for further research on dosing-interval optimization, and present chances for accelerated population coverage and vaccine equity.

### 5.3. Mix-and-Match Vaccination

With an exception for the Janssen vaccine, the majority of the COVID-19 vaccines require a two-dose prime-boost immunization strategy as the primary-course dosing, administered between 3 and 12 weeks apart (Table 1). Traditionally, most COVID-19 vaccines follow a homologous prime-boost schedule: both the priming and booster doses utilize the same vaccine type. Reports of the rare yet potentially life-threatening occurrence of thromboembolic events following adenovirus vector-based vaccine administration (the Oxford/AstraZeneca and Janssen vaccines) resulted in modifications of key vaccination policies in multiple European countries, leading to the introduction of heterologous prime-boost strategies [72]. Individuals who received an adenovirus vector vaccine as a primer are now recommended for a booster with an alternative vaccine type, such as the mRNA vaccines. A Phase I/II trial in the United States evaluating the prime-boost permutations of the Moderna, Janssen, and Pfizer/BioNTech vaccines found that both heterologous regimens elicited higher antibody titers (from 6.2 to 76 times), compared with a 4.2–20-time increment following a homologous schedule [73]. Similar systemic reactogenicity profiles were also noted. In addition, large nationwide cohort studies observed a comparable vaccine efficacy against COVID-19 disease after a heterologous AstraZeneca/mRNA-based vaccine prime-boost (79–88%), as opposed to after a homologous mRNA vaccination (80–90%) [74].

The international interest in the use of a heterologous prime-boost COVID-19 vaccine regimen with similar immunogenicity and reactogenicity profiles is growing, and particularly to help solve the problem of fluctuating vaccine stockpiles and, hence, the delayed administration of booster doses. Due to the aforementioned justifications, the Centers for Disease Control and Prevention (CDC) Advisory Committee on Immunization Practices (ACIP) supported the mix-and-match heterologous immunization strategy for COVID-19 through its guidance release in October 2021 [75].

## 6. Vaccine Acceptance and Safety

### 6.1. The Issue of Vaccine Hesitancy

Ensuring widespread COVID-19 vaccine acceptance is just as important as the strive towards equitable vaccine access. Vaccine hesitancy is a significant hurdle to achieving sufficient COVID-19 immunization coverage, as it is influenced by the trust in vaccines and the institutions that administer them. A systematic review and meta-analysis of 172 studies across 50 countries reported a pooled acceptance proportion of 61% (CI: 59–64%). Higher acceptance was found in the Southeast Asian countries, in males, among healthcare professionals, and for vaccines with 95% effectiveness [76]. Lazarus et al. (2021) suggested that Asian countries leaned towards a relatively high acceptance rate, which is attributable to unwavering support for its community, the social support system, and the trust placed in their central governments [77]. Another study observed a higher willingness for COVID-19 uptake in LMIC samples across Asia, Africa, South Asia, and Latin America (80.3%, 95% CI: 74.9–85.6%), compared with high-income nations, such as the United States (64.6%, CI: 61.8–67.3) and Russia (30.4%, CI: 29.1–31.7%) [78]. Such wide differences in the willingness to receive COVID-19 vaccination are alarming and need to be addressed to prevent variations in international immunization coverage.

### 6.2. Rarity of Severe Adverse Events following COVID-19 Vaccination

Concerns about the safety and possible side effects have frequently been cited as barriers to COVID-19 vaccine uptake [78]. Of note, the primary severe adverse events reported among adults have been extremely rare: anaphylaxis (from 2.5 to 4.8 cases per million doses) and myocarditis (from 6 to 27 cases per million) for mRNA-typed vaccines. Reports have also been received on Guillain–Barré syndrome (7.8 cases per million) for the Janssen vaccine, and thrombosis with thrombocytopenia syndrome (TTS) for both the Oxford/AstraZeneca vaccine (2 cases per million) and Janssen vaccine (3 cases per million) [79]. Surveillance systems have reported no obvious signals of adverse maternal, pregnancy, or infant outcomes following mRNA vaccination [80]. In contrast, when unvaccinated, an elevated risk for severe COVID-19 infections has been observed in pregnancy, with the babies more likely to be admitted to the neonatal intensive care unit [81]. Therefore, the delivery of accurate information through widespread media coverage on the benefits of vaccines and the rarity of potential serious adverse events may alleviate the problem of vaccine hesitancy.

## 7. Conclusions

The launch of global COVID-19 vaccination campaigns has generated hope that the global population and societies around the world stand a chance of returning to a certain degree of normalcy. However, the COVID-19 vaccines that are safe and effective can only be meaningful to the global community if they are accessible in a timely manner. Despite the extraordinary achievement of delivering vaccine candidates to the wider market within a short time span of one year, the multiple challenges faced to achieve this, persist to this day. A collaborative response to achieve global equitable access is key. In this paper, we discuss the global vaccination challenges that continue to maneuver the direction of vaccine development, manufacture, allocation, and deployment. Decisionmakers around the world need to remain vigilant and responsive to their commitment to ending the pandemic and facilitating the transition to endemicity. These unprecedented times call for commitments through global unity to render vaccines equitable. Indeed, it is an ethical test that transcends borders.

## Figures and Tables

**Figure 1 vaccines-10-01306-f001:**
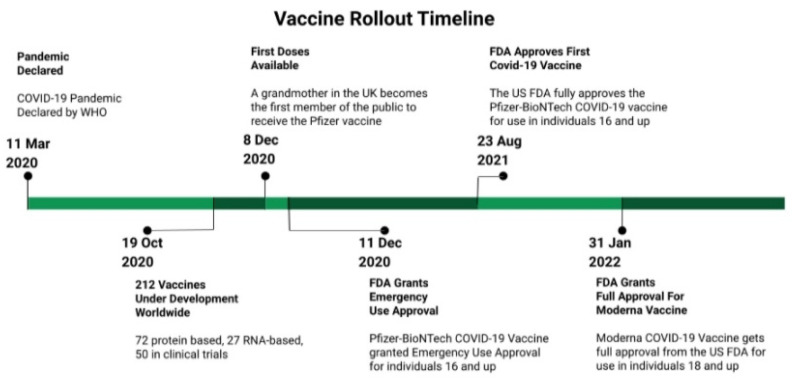
A brief timeline of selected events related to the vaccine rollout. Adapted with permission from Peterson et al. [6]. 2022, Christopher J. Peterson.

**Table 1 vaccines-10-01306-t001:** Comparison of COVID-19 Vaccines’ Attributes and Storage and Handling Requirements as of 1 July 2022.

	Pfizer/BioNTech			Moderna [58]	Oxford/AstraZeneca [59]	Janssen [43]	CanSino [60]
	Formulation for from 5 to 11 Years of Age (Orange Cap) [61]	Formulation for 12 Years of Age and Older (Purple Cap) [62]	Formulation for 12 Years of Age and Older (Gray Cap) [63]				
Research name	BNT162b2			mRNA-1273	AZD1222; ChAdOx1 nCoV-19	Ad26.CoV2.S	Ad5-nCoV
Vaccine type	mRNA (nucleic acid)			mRNA (nucleic acid)	Viral vector	Viral vector	Viral vector
Country of origin	United States/Germany			United States	United Kingdom	United States	China
Primary-course dosing	2 doses 3 weeks apart	2 doses 3 weeks apart	2 doses 3 weeks apart	2 doses 4 weeks apart	2 doses from 4 to 12 weeks apart	Single dose	Single dose
Efficacy (symptomatic COVID-19), % (95% CI)	95.0% (90.3–97.6)			94.1% (89.3–96.8)	76.0% (68.0–82.0)	72.0% (58.2–81.7)	65.28%
Efficacy against infection with COVID-19 variant	B.1.1.7 (α): 75–95% B.1.351 (β): 75% B.1.617.2 (δ): 42–79% B.1.1.529 (O): 65.5%			B.1.1.7 (α): 84–99% B.1.351 (β): 96% B.1.617.2 (δ): 76–84% B.1.1.529 (O): 75.1%	B.1.1.7 (α): 79% B.1.617.2 (δ): 60–67% B.1.1.529 (O): 48.9%	No data	B.1.617.2 (δ): 61.5%
Dose volume	0.2 mL	0.3 mL	0.3 mL	0.5 mL	0.5 mL	0.5 mL	0.5 mL
Doses per vial	10	6	6	10	10	10	Single-dose vial and three-dose vial
Dilution required	Yes (1.3 mL diluent)	Yes (1.8 mL diluent)	No	No	No	No	No
Storage temperature for unopened vials	From −90 °C to −60 °C (until the expiration date); from 2 °C to 8 °C (10 weeks)	From −90 °C to −60 °C (until the expiration date); from −25 °C to −15 °C (2 weeks); from 2 °C to 8 °C (31 days)	From −90 °C to −60 °C (until the expiration date); from 2 °C to 8 °C (10 weeks)	From −50 °C to −15 °C (during shipping); from 2 °C to 8 °C (up to 30 days)	From 2 °C to 8 °C (6 months)	From 2 °C to 8 °C (until the expiration date)	From 2 °C to 8 °C (6 months)
Recommendation for predrawn doses	6 h (from 2 °C to 25 °C) (56)	6 h (from 2 °C to 25 °C) (8)	6 h (from 2 °C to 25 °C) (8)	6 h (from 2 °C to 25 °C) (56)	1 h (up to 30 °C); 6 h (from 2 °C to 8 °C) (57)	No data	No data
Recommendation for storing open vials	12 h post dilution (from 2 °C to 25 °C)	6 h post dilution (from 2 °C to 25 °C)	12 h of first puncture (from 2 °C to 25 °C)	12 h (from 2 °C to 25 °C), or after the vial has been punctured 20 times	6 h (from 2 °C to 25 °C)	2 h (up to 25 °C); 6 h (from 2 °C to 8 °C)	Opened multidose vial: 6 h (from 2 °C to 8 °C)
Transport limitations	Alternative temperature from 2 °C to 8 °C is allowed	12 h thawed from 2 °C to 8 °C	Alternative temperature from 2 °C to 8 °C is allowed	12 h thawed from 2 °C to 8 °C	Nil	Nil	Nil

## Data Availability

All relevant data are available in the article.

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
