# Peer review of "The Race for Global Equitable Access to COVID-19 Vaccines"

_vaccines, 2022, doi:10.3390/vaccines10081306_

Round 1

Reviewer 1 Report

The review by Khairi et al., summarizes the current updates on COVID vaccine development and the global efforts to make the vaccines available for public. The review also talks about the limitations of certain vaccine candidates based on their storage, and dosing requirements, and authors comment on the measures being taken to increase the production, allocation, and deployment of COVID vaccines. 

I have the following comments- 

1. I believe that the introduction can be elaborated a bit more. You can comment on the fact that mRNA vaccines for COVID are the first to ever receive FDA approval and the accelerated rate at which this occurred, and how it impacted the pandemic. 

2. Although it is important to ensure that vaccines are accessible to everyone, their safety and efficacy are utmost important. There have been many complications with certain vaccines including both single-dose and multi-dose. Please address this area as well in the review. 

3. Under the Development of single-dose covid vaccines section, please discuss the effectiveness of the current available single-dose vaccines. 

4. Under section 8, please add any effectiveness data available from subjects that received combination vaccines. 

5. Line 50- Please add "to control the spread" next to "consolidated approach". 

Reviewer 2 Report

A short presentation of COVID-19 pathology would be welcomed in the Introduction section

Diverse COVID-19 variants should be discussed, efficiency of vaccines against certain subtypes of the virus should be presented

The timeline of vaccine introduction is missing

Table 1 is not formatted according to journal recommendations

Issues regarding reluctance at vaccination is a reality in several countries, however it is not discussed in the article

Regulatory data regarding COVID vaccine introductions should be presented
